# Interplay of RNA-Binding Proteins and microRNAs in Neurodegenerative Diseases

**DOI:** 10.3390/ijms22105292

**Published:** 2021-05-18

**Authors:** Chisato Kinoshita, Noriko Kubota, Koji Aoyama

**Affiliations:** 1Department of Pharmacology, Teikyo University School of Medicine, 2-11-1 Kaga, Itabashi, Tokyo 173-8605, Japan; kubota3724@gmail.com; 2Teikyo University Support Center for Women Physicians and Researchers, 2-11-1 Kaga, Itabashi, Tokyo 173-8605, Japan

**Keywords:** neurodegenerative disease, RNA-binding protein, microRNA

## Abstract

The number of patients with neurodegenerative diseases (NDs) is increasing, along with the growing number of older adults. This escalation threatens to create a medical and social crisis. NDs include a large spectrum of heterogeneous and multifactorial pathologies, such as amyotrophic lateral sclerosis, frontotemporal dementia, Alzheimer’s disease, Parkinson’s disease, Huntington’s disease and multiple system atrophy, and the formation of inclusion bodies resulting from protein misfolding and aggregation is a hallmark of these disorders. The proteinaceous components of the pathological inclusions include several RNA-binding proteins (RBPs), which play important roles in splicing, stability, transcription and translation. In addition, RBPs were shown to play a critical role in regulating miRNA biogenesis and metabolism. The dysfunction of both RBPs and miRNAs is often observed in several NDs. Thus, the data about the interplay among RBPs and miRNAs and their cooperation in brain functions would be important to know for better understanding NDs and the development of effective therapeutics. In this review, we focused on the connection between miRNAs, RBPs and neurodegenerative diseases.

## 1. Introduction

Neurodegenerative diseases (NDs) are progressive and fatal diseases characterized by selective neuronal vulnerability with degeneration in specific brain regions, and deposits of abnormal proteins are often found in the brains of ND patients [1]. NDs are age-dependent disorders and have recently become a major medical and social threat, in part due to the growing population of elderly people in super-aged societies worldwide. These disorders include Alzheimer’s disease (AD), Parkinson’s disease (PD), Huntington’s disease (HD), amyotrophic lateral sclerosis (ALS), frontotemporal dementia (FTD) and multiple system atrophy (MSA). Their symptoms partly overlap, with some causing memory and cognitive impairments and others affecting a person’s ability to move, speak and breathe. NDs impose a major burden on societies, families and individuals, not only through the loss of labor but, also, through the frequent requirement of 24-h support by the relatives of the affected individual. The development of therapeutics effective for NDs is absolutely crucial to solving this problem, and to this end, it is necessary to understand the causes and mechanisms of each disease.

Recent lines of evidence show that several kinds of inclusion bodies are hallmarks of NDs, and these inclusion bodies contain several RNA-binding proteins (RBPs). More than 2000 RBPs have been identified, with the recent development of high-throughput technologies, such as next-generation sequencing and crosslinking immunoprecipitation sequencing, greatly facilitating the search and permitting investigation of the interactions between RBPs and their target RNAs in a quantitative and high-resolution manner [2]. RBPs recognize consensus sequences and regulate gene expression in multiple biological events, such as alternative splicing, and RNA biogenesis, maturation, stability, localization, transport and translation efficiency [3]. The diversity of RBPs may have been evolutionally increased with the increase in the number of introns [4]. Such diversity could enable eukaryotes to utilize exons in various arrangements, giving rise to specificity with respect to the developmental stages, as well as tissues and cells. The specific expression of RBPs with appropriate timing and localization could contribute to the precise machinery of RNA synthesis and/or metabolism. Conversely, the dysregulation of RBPs by abnormal modifications or genetic mutations can be either a primary cause of disease or a consequence of NDs [5].

RNA is rarely alone and is not surrounded by a lipid membrane. Generally, RNA localizations are controlled by binding with several RBPs that form ribonucleoprotein (RNP) complexes [6]. These membrane-less compartments are called RNA granules, a category that includes processing bodies (P-bodies) and stress granules [7]. RNA granules form via liquid–liquid phase separation, which is driven by a dynamic network of multivalent interactions of proteins containing intrinsically disordered regions and low complexity domains. P-bodies are cytoplasmic RNA granules, which are involved in translational repression and/or RNA decay [8]. Under cellular stress conditions, stress granules heterogeneously assemble in the cytoplasm to mediate the pro-survival adaptive responses and, subsequently, disassemble soon after the stress is removed [9].

Several NDs are thought to proceed through the aggregation of misfolded proteins because of genetic mutation, and/or their dysregulation promotes the assembly of stress granules and triggers their transition to chronic pathological inclusions [5]. Several mutated RBPs have been found to be linked to the NDs, and these RBPs generally have structurally unstable regions that play critical roles in protein misfolding [10].

MicroRNAs (miRNAs) are a class of RNA molecules with roles in the post-transcriptional modulation of gene expression [11]. Currently, 38,589 hairpin precursors and 48,860 mature miRNAs from 271 organisms are registered in the miRBase database (release 22). The function of miRNAs has been clarified to be precisely organized by a large number of RBPs, as RBPs control different stages of miRNA biogenesis and their localization, degradation and activity [12]. Some of the RBPs play central roles in the functional machinery of miRNAs across multiple cells, and others may contribute tissue- and/or developmental stage-specificity [11]. Interestingly, miRNA studies in humans revealed that the down-regulation or overexpression of miRNAs regulated by RBPs is linked to clinically important diseases, including NDs [13]. These facts are consistent with the observation that dysregulations of RBPs and miRNAs are simultaneously detected in ND patients. In this review, we will focus on the cooperative or competitive interactions between RBPs and miRNAs in the brain and their disruption in NDs.

## 2. RNA-Binding Proteins (RBPs)

RBPs are a specific set of proteins that can bind to double- or single-stranded RNA [3]. Classically, RBPs participate in the formation of RNP complexes, which are involved in post-transcriptional gene regulation [14]. RBPs play a crucial role in regulating all aspects of mRNA biogenesis and metabolism, including splicing, editing, polyadenylation, nuclear export, stabilization, localization and translation [15]. Most RBPs typically possess one or multiple RNA-binding domains (RBDs), such as RNA recognition motifs (RRMs), zinc finger domains (ZF domains), hnRNP K homology domains (KH domains) and so on [16]. Although numerous RBPs have been discovered and investigated over the years, how RBPs interact with and affect RNA has not been fully elucidated, and relatively few RBPs have been studied systematically [17]. In the past decade, proteome-wide studies using large-scale quantitative methods such as next-generation sequencing and modern mass spectrometry have dramatically increased the number of proteins implicated as RBPs, some of which surprisingly lack conventional RBDs [18,19,20,21]. In this section, we will review the typical RBPs related to the NDs reviewed in Section 4, which we roughly categorized by their generic names (Table 1). Note that the accurate and precise classification of RBPs is extremely difficult, since there are numerous determinants for RBP features, such as conserved domains, protein structures, formed molecular complexes, biological functions and related diseases. 

### 2.1. RNA-Binding Motif Proteins (RBM)

RBMs belong to a huge and diverse family of glycine-rich proteins containing at least one RRM protein domain [25]. According to a search by the HUGO Gene Nomenclature Committee, 52 RBM genes have been identified so far. RBM genes are basically designated by the prefix RBM, followed by a numeric and/or alphabetic suffix (e.g., RBMX, RBM8A and RBM45). Each RBM protein has an important role in biological events such as pre-mRNA splicing, posttranscriptional regulation, mRNA stability, protein synthesis and so on [26,27,28]. Emerging studies have shown that RBM is involved in hyperthermia, cancer, apoptosis and neuroprotection [28,29,30,31,32].

### 2.2. Heterogeneous Nuclear Ribonucleoprotein (hnRNP)

hnRNPs are a set of primarily nuclear proteins that assist in the control of maturation of heterogeneous nuclear RNAs (hnRNAs/pre-mRNAs) into messenger RNAs (mRNAs), stabilize mRNAs during their cellular transport and control the translation of mRNAs [33,34]. The immunopurification of hnRNP complexes using monoclonal antibodies against hnRNP C in combination with two-dimensional gel electrophoresis revealed a collection of 20 proteins that were named hnRNPs A–U, which have different molecular weights ranging from 34 to 120 kDa [25]. Four unique RBDs were identified in hnRNPs: the RRM, the quasi-RRM (qRRM), a glycine-rich domain constituting an arginine-glycine-glycine (RGG) box and a KH domain. Some of the hnRNPs have alternative names; hnRNP D is also known as the ARE/poly(U)-binding/degradation factor 1 (AUF1), hnRNP G is alternatively named RNA-binding motif protein chromosome X (RBMX), hnRNP I is known as Polypyrimidine tract-binding protein 1 (PTBP1) and hnRNP P2 is better known as the Fused in sarcoma/Translated in liposarcoma (FUS/TLS) (see Section 2.4). In addition, the trans-active response DNA-binding protein of 43 kDa (TDP-43) is known to be a member of the hnRNP family (see Section 2.3).

### 2.3. Trans-Active Response DNA-Binding Protein of 43 kDa (TDP-43)

TDP-43 was identified in 1995 as a repressor protein associated with human immunodeficiency virus type 1 (HIV-1) transcription. TDP-43 binds specifically to pyrimidine-rich DNA motifs in a long terminal repeat known as the trans-active response (TAR) element, which is critical for regulation of the viral gene expression [35]. Subsequent studies have shown that TDP-43 is a highly conserved and ubiquitously expressed RNA/DNA-binding protein that belongs to the hnRNP family [36]. TDP-43 regulates mRNA splicing, stability and translation, as well as gene transcription, through binding to RNA or DNA [37,38,39]. TDP-43 has been implicated as an essential regulator of transcriptional events in the central nervous system (CNS), and genetic mutations of TDP-43 and aggregation of the TDP-43 protein were identified in patients with NDs [40].

### 2.4. FET Proteins

FUS/TLS, Ewing sarcoma (EWS: also known as Ewing sarcoma breakpoint region 1 (EWSR1)) and TATA-binding associated factor 15 (TAF15) belong to the FET family of DNA and RNA-binding proteins [41]. These three proteins are structurally similar and ubiquitously expressed and were first discovered upon the characterization of fusion oncogenes in human sarcomas and leukemias [42,43,44]. They mainly contain several conserved domains: a N-terminal serine-tyrosine-glycine-glutamine (SYGQ) domain that acts as a transcriptional activation domain, a central conserved RRM and a C-terminal ZF domain involved in RNA and DNA binding, respectively, and multiple RGG-rich regions in the C-terminal region that affect RNA binding [45]. It has been suggested that FET proteins are involved in neurological diseases, such as frontotemporal lobar degeneration (FTLD) and ALS, where they have been found in cytoplasmic aggregates [41].

### 2.5. Embryonic Lethal/Abnormal Visual System (ELAV)/Hu Family

ELAV/Hu family proteins are important regulators of neuronal RNA processing and have been implicated in alternative splicing, polyadenylation, mRNA localization, mRNA turnover and translation [46]. They have a variable N-terminal extension, three RRMs and a hinge region separating the first two RRMs from the third. There are four ELAV/Hu family members in mammals: HuR (also termed HuA), HuB, HuC and HuD [47]. Whereas HuR is expressed ubiquitously throughout the organism, the other three family members (HuB, HuC and HuD) are exclusively neuronal due to translational repression in other cell types [47].

### 2.6. CUG-BP, ELAV-Like Family (CELF) Proteins

CELF regulates several steps of RNA processing in the nucleus and cytoplasm, including pre-mRNA alternative splicing, C to U RNA editing, deadenylation, mRNA decay and translation [48]. CELF proteins have three conserved RRMs, RRM1 and RRM2 in the N-terminus of the protein and RRM3 in the C-terminus [49]. Six members of CELF have been identified so far, and they can be divided into two distinct subfamilies based on their phylogeny: CELF1-2 and CELF3-6 [50]. CELF proteins have been shown to regulate the alternative splicing of transcripts involved in neuronal functions, and the dysregulation of CELF-mediated alternative splicing in the brain has been implicated in the pathogenesis of NDs [51].

### 2.7. Regulator of Calcineurin (RCAN)

RCAN was first reported as Down syndrome critical region 1 (DSCR1), which is encoded in a region termed the Down syndrome critical region (DSCR); DSCR is located on human chromosome 21 and was thought to participate in the onset of Down syndrome at the time [52]. There are three members of the RCAN gene family, RCAN1, RCAN2 and RCAN3, present in Gnathostomata [53]. Several well-conserved domains within RCAN structure have been identified, i.e., the N-terminal RRM domain, LxxP motif, PxIxIT-like motif and TxxP motif [54,55]. One of the well-characterized functions of RCANs is to inhibit the activity of calcineurin, which is a calcium/calmodulin-activated serine-threonine phosphatase; through this function, RCANs play an important role as a modulator of the calcineurin–nuclear factor of activated T cells (NFAT)-signaling pathway [56]. Recent lines of evidence have shown that RCAN is a multifunctional protein involved in the regulation of neurons, inflammation, protein glycosylation mitochondria homeostasis, RNA binding, circadian rhythms, obesity and thermogenesis, and some of these activities are calcineurin-independent [53].

### 2.8. Musashi (Msi)

Msi was originally discovered in Drosophila and named after the iconic Japanese samurai Miyamoto Musashi, because the double-bristle phenotype of mutant Drosophila was reminiscent of his fighting style with two swords [57]. Msi is an evolutionarily conserved family of RBPs that is preferentially expressed in the nervous system [58]. In vertebrates, two homologous Msi genes have been identified, Msi1 and Msi2 [59]. Each Msi protein contains two N-terminal RRMs (RRM1 and RRM2) that mediate the binding to their target mRNAs [60]. Both proteins play an essential role in stem cell self-renewal and maintenance across a variety of different biological systems [61,62]. Recent lines of evidence have shown that any dysregulation of Msi1 and Msi2 can lead to cellular dysfunctions promoting tissue instability and tumorigenesis [63].

### 2.9. Fragile X Mental Retardation Protein (FMRP) 

Fragile X syndrome is the leading inherited form of intellectual disability and autism spectrum disorder, and patients can present with severe behavioral alterations, including hyperactivity, impulsivity and anxiety, in addition to poor language development and seizures. This disease is caused by the loss of the FMRP [64]. FMRP is an RBP encoded by the FMR1 gene that regulates mRNA translation, localization and stability, thereby playing a critical role in RNA metabolism, neuronal plasticity and muscle development [65]. FMRP consists of three major RBDs, i.e., two KH domains and an RGG box domain [66,67]. FMRP is part of a larger family of RBPs known as Fragile X-related proteins (FXRs), which includes FXR1P and FXR2P [68]. All three FXRs can regulate adult neurogenesis in neuroplasticity, learning and cognition, although most studies have focused only on FMRP [69].

### 2.10. T-Cell Restricted Intracellular Antigen (TIA) Proteins

TIA functions as a posttranscriptional regulator of gene expression by binding to cis elements found in untranslated regions of selected mRNAs to participate in translational repression in response to various stresses, as well as in the regulation of alternative splicing of target mRNAs [70]. Two isoforms of TIA proteins, TIA1 and TIA1-related protein (TIAR), have been identified and shown to possess three N-terminal RRMs and a C-terminal glutamine-rich prion-like domain (PrLD) [70,71]. They are ubiquitously but predominantly expressed in the brain, testis and spleen [72]. Recent lines of evidence showed that TIA proteins are associated with pathological inflammation or aggregate formation [70].

### 2.11. Tristetraprolin (TTP)

The TTP family includes Zinc finger protein 36 (ZFP36; commonly referred to as TTP), ZFP36-like 1 (ZFP36L1) and ZFP36-like 2 (ZFP36L2) [73]. These RBPs are characterized by the presence of one or more ZF domain(s) that contain three cysteine residues and one histidine residue [74]. Mechanistically, they function by binding to the AU-rich elements within the 3′-untranslated regions (UTRs) of their target mRNAs in a sequence- and structure-specific manner through a highly conserved ZF domain and catalyzing the removal of the poly(A) tail, thus resulting in their mRNA decay [73,74,75].

### 2.12. Serine/Arginine-Rich Splicing Factor (SRSF)

SRSFs belong to the serine arginine-rich protein (SR protein) family and are highly conserved in plants and animals [76]. Canonically, they consist of 12 members in mammals (SRSF1-12), which have very similar domains characterized by the presence of one or two N-terminal RRMs and a C-terminal domain enriched with arginine and serine amino acid sequences (RS domain) [76,77]. SR proteins play a pivotal role in almost all processes of RNA metabolism, including both constitutive and alternative splicing, through binding to exonic splicing enhancers (ESE) and recruiting small nuclear RNP (snRNP) and its cofactors to the splicing sites [76,78]. 

### 2.13. Neuro-Oncological Ventral Antigen (NOVA)

NOVA was initially identified as an antigen in a rare neurological disorder known as paraneoplastic opsoclonus-myoclonus ataxia [79]. The NOVA family is composed of two members, NOVA1 and NOVA2, both of which have three conserved KH domains and have been demonstrated to regulate numerous alternatively spliced exons through binding to the target pre-mRNAs [80]. In addition, NOVA1 and NOVA2 have been shown to be brain-specific splicing factors that interact with RNA-containing repeats of the YCAY (Y = C or U) sequence [81].

### 2.14. Matrin3

Matrin3 is an RNA-binding protein named for its localization in the nuclear matrix, and it has various roles in RNA metabolism, including alternative splicing [82,83,84]. Matrin3 preferentially binds pyrimidine-rich sequences, including the polypyrimidine tract [84]. The putative domain structures of Matrin3 contain two ZF domains, two RRMs, an N-terminal nuclear export signal and a C-terminal nuclear localization signal. Matrin3 was originally identified as a nuclear matrix protein and more recently recognized as an ALS-causing gene [85,86,87].

### 2.15. Poly(A)-Binding Protein (PABP)

PABP was originally isolated from a specific nucleoprotein complex found in the poly(A) tail [88] and was shown to stimulate the translation of mRNAs [89]. In humans, five PABP proteins have been identified: the cytoplasmic PABPs (PABPC1, PABPC3 and iPABP); nuclear PABP (PABPN1) and X-linked PABP (PABPC5). Four of the five PABPs have a similar structure that includes four RRMs, whereas PABPN1 has a single RRM [90,91]. The small alanine expansions in the PABPN1 protein are known to cause oculopharyngeal muscular dystrophy, resulting in the intranuclear accumulation of PABPN1 in skeletal muscle [92].

### 2.16. Scaffold Attachment Factor B (SAFB)

SAFB was first identified as a DNA-binding protein that specifically binds to scaffold or matrix attachment region DNA elements (S/MAR DNA) and was recently reported to also bind to RNA [93]. The SAFB proteins play crucial roles in transcription, mRNA processing, DNA repair and the formation of transcriptosomal complexes. All three SAFB proteins, SAFB1, SAFB2 and the SAFB-like transcriptional modulator, SLTM, have a well-conserved DNA-binding (SAF-A/B, Acinus and PIAS, SAP) domain; an RBD and an arginine/glycine motif-RGG/RG domain. The SAFB proteins are widely expressed, with particularly high expression of SAFB1 and SAFB2 in the human central nervous and immune systems.

### 2.17. Sm Proteins—Components of Small Nuclear Ribonucleoprotein (snRNP)

snRNP particles are RNA–protein complexes that combine with unmodified pre-mRNA and various other nuclear proteins to form a spliceosome [94]. They are essential in several aspects of nuclear posttranscriptional gene expression, including pre-mRNA splicing, telescripting and 3′ end processing of histone pre-mRNAs [95]. Each snRNP in eukaryotes consists of specific uridine-rich snRNAs, which are designated by the prefix U followed by a number and a specific set of proteins—e.g., U1snRNP. Most snRNPs have an Sm core, a heptameric ring of Sm proteins consisting of B/B′, D1, D2, D3, E, F and G that surrounds an RNA sequence element called the Sm site, a doughnut-shaped core RNP structure [96]. In addition, U1snRNP has the specific proteins U1-70K, A and C. U2snRNP has two unique proteins, U2-A′ and B′′. U4/U6 has a unique 120–140-kDa protein, and U5snRNP has eight unique proteins, including 100-, 102- and 200-kDa doublets, in addition to the Sm core particle.

### 2.18. Ribosomal Proteins

Ribosomal proteins are known to play an essential role in ribosome assembly and protein translation and are also involved in various physiological and pathological processes [97]. Ribosomes in eukaryotes contain four rRNAs and around 80 ribosomal proteins [98,99]. Ribosomal proteins from the small and large subunits are basically named RPS and RPL with a suffix number, respectively. The naming of ribosomal proteins has been complicated by the increasing number of ribosomal proteins identified in various species, resulting in a chaotic situation in which different research groups adopt their own naming systems; a new nomenclature has been proposed to resolve this problem [100].

## 3. Involvement of RNA-Binding Proteins in Neuronal microRNA Biogenesis

Increasing lines of evidence suggests that functional interactions and cooperation between RBPs and miRNAs are a key mechanism in the post-transcriptional regulation of gene expression [101]. miRNAs are a class of short noncoding RNAs that are approximately twenty nucleotides in length [11]. Their function consists mostly of silencing target expressions by binding to target gene transcripts located mainly at the 3’-UTR. Most miRNA genes are located in intergenic regions or in an antisense orientation to gene regions on the genome. Clustered miRNAs can either be simultaneously transcribed from single polycistronic transcripts containing multiple miRNAs or independently transcribed. The biogenesis of miRNAs is precisely regulated by molecular complexes including RBPs at almost every step [102] (Figure 1). The primary miRNAs (pri-miRNAs)—that are primary transcripts containing stem-loop structures and are usually thousands of nucleotides in length—are initially transcribed by a polymerase—in most cases, RNA polymerase II. Then, the pri-miRNAs are cleaved by a complex called a Microprocessor, which contains the ribonuclease III Drosha and the RNA-binding protein DGCR8/Pasha. The Microprocessor generates small hairpin-shaped RNAs of approximately 70–100 nucleotides in length called miRNA precursors (pre-miRNAs). This step has been reported to involve NDs-associated RBPs such as TDP-43, FET proteins and hnRNPs. TDP-43 associates with the Drosha complex and binds to a set of pri-miRNAs, such as pri-miR-132, -143, -558 and -574, and promotes their cleavage [103]. FUS interacts with specific neuronal pri-miRNA sequences, together with Drosha, and assists in recruiting Drosha to the chromatin, allowing the efficient processing of neuronal miRNAs such as pri-miR-9, -125b and -132 [104]. In addition, the specific and prevalent interactions between EWS and Drosha with a large subset of pri-miRNAs such as pri-miR-29b, -18b, -34a, -222 and -let7 were shown to enhance the recruitment of Drosha to chromatin [105,106]. The HuR-mediated binding of Msi2 to the conserved terminal loop of pri-miR-7 also inhibits Drosha cleavage [107]. Moreover, hnRNP A1 binding to pri-let-7 leads to the inhibition of Drosha processing, whereas binding to pri-miR-18a contributes to the stimulation of processing [108,109,110,111]. In the next step, pre-miRNAs exported by exportin-5 in complex with RAN-GTP are processed by a double-stranded ribonuclease III enzyme, termed Dicer, which is complexed with a double-stranded RNA-binding protein, TRBP. AUF1 has a role in suppressing miRNA production by reducing Dicer production in association with the endogenous Dicer mRNA [112]. TDP-43 is also involved in this step by facilitating processing through the binding of Dicer to specific sites of pre-miRNAs such as pre-miR-143 and -574 [103]. Next, an Argonaute protein forms a part of an effector complex, called the RNA-induced silencing complex (RISC), onto which the mature miRNA duplexes are loaded. TDP-43 is negatively involved in the step of loading specific miRNAs such as miR-1 and -206 onto the RISC complex [113]. Conversely, the complex formed by the association of PABP with mRNA promotes miRISC recruitment [114]. Finally, one strand of the miRNA is removed from RISC to generate the mature RISC that induces gene silencing. The post-transcriptional regulation by the RISC complex is mediated by the incomplete base pairing of miRNA–mRNA interactions, likely due to the targeting of multiple transcripts, which contributes to the complexity or redundancy of miRNA systems.

## 4. Interplay between RNA-Binding Proteins and microRNAs in Neurodegenerative Disease

The progressive degeneration of the structure and function of the nervous system is the main feature of NDs such as AD, PD, ALS, FTD, HD and MSA [1]. These diseases primarily occur in the later stages of life. As an ironic side effect of the worldwide increases in life expectancy and resulting growth in the aging population, the number of people suffering from NDs has also grown. Currently, no NDs are curable; the available treatments only manage the symptoms or halt the disease progression.

The genetic mutations that cause the pathogenesis of familial NDs have been well-studied; however, with the exception of HD, the majority of NDs are sporadic. Our knowledge of sporadic NDs is unfortunately not sufficient to understand their molecular etiologies, even though their pathologies are well-established. Several lines of evidence indicate that the onset of sporadic diseases is due to the synergistic effect of genetic polymorphism, environmental factors or their combination with overlapping disease mechanisms, as seen in the familial disease pathologies [115]. In fact, gene expression profiling from clinical studies has revealed a profound dysregulation of gene expression in the tissues of patients with NDs [116], suggesting that the post-transcriptional and/or post-translational regulation of the gene expression would clearly affect the onset or progression of NDs.

There is accumulating evidence that the dysregulation of the post-transcriptional molecular architecture composed of several types of RNA and protein causes NDs. Here, we focused on the regulatory mechanism of miRNAs and RBPs in NDs.

### 4.1. Amyotrophic Lateral Sclerosis (ALS) and Frontotemporal Dementia (FTD)

ALS and FTD have been perhaps the most studied of the diseases associated with RBPs, since numerous genetic mutations in patients with ALS and FTD have been discovered in genes encoding RBPs [10].

ALS is also known as Lou Gehrig’s disease, which is a chronic progressive disease characterized by the selective degeneration of motor neurons in the spinal cord and motor cortex, normally causing death within 3–5 years of onset [117]. The most common genetic cause of ALS is a GGGGCC hexanucleotide repeat expansion in the first intron of the C9orf72 gene. Mutations in several genes, including TDP-43, FET proteins, hnRNP A1, hnRNP A2B1 and Matrin-3, have also been identified as potential genetic risk factors [10]. 

FTD is a heterogeneous clinical syndrome associated with FTLD characterized by a progressive decline in behavior or language associated with degeneration of the frontal and anterior temporal lobes [118]. The classification of FTLD is based on the aggregated proteins in the inclusions, which are tau-positive (FTLD-tau), TDP-43-positive (FTLD-TDP) or FUS-positive (FTLD-FUS) [119]. FTLD overlaps clinically and pathologically with the atypical parkinsonian disorders, including corticobasal degeneration and progressive supranuclear palsy, as well as with ALS [120].

Although TDP-43 regulates splicing, stability, translation and transcription by binding to both RNA and DNA, its aberrant phase transitions, splicing dysfunction and dysregulation of its nuclear-cytoplasmic shuttling are neurotoxic and thereby induce proteinopathy in ALS and FTLD-TDP [121,122,123]. TDP-43 has been reported to negatively affect the Drosha and/or Dicer processing of miR-181c-5p and miR-27b-3p, which, in turn, has a negative effect on TDP-43 expression [124]. Both miRNAs have been shown to be reduced in the spinal cord of patients with ALS [125]. Further, miR-NID1 (Neurexin-1 (NRXN1) intron-derived miRNA, which corresponds to miR-8485) represses the expression of NRXN1 by binding to TDP-43 in neuroblastoma cell lines [126]. FUS mis-localization facilitates its toxic aggregation of FUS by disrupting the interaction with ALS-associated RBPs such as TDP-43, EWS, TAF15, Matrin3, hnRNP A1 and hnRNP A2B1 [127,128]. Two ALS-associated RBPs, FUS and TDP-43, are involved in the regulation of miR-183/96/182 biogenesis and memory suppressor protein phosphatase 1, and their alteration results in age-related memory decline in the pathogenesis of ALS and FTLD [129]. TAF15, the function of which resembles the function of FUS, is important for miRNA processing, as well as alternative splicing events in neurons, and its dysregulation is relevant to ALS, although the effects of TAF15 on these events are regarded as less effective than the effects of FUS [128,130,131]. TAF15 plays a role in the biosynthesis of miRNAs produced by the miR-17-92 cluster, thereby regulating the genes related to cell proliferation in human neuroblastoma cell lines [132]. Dysregulation of the miR-17-92 cluster has been shown to contribute to the particular susceptibility of motor neurons in ALS model mice [133]. The cytoplasmic expression of hnRNP A1 has been reported to be increased in response to excitotoxic stress in primary cortical neurons, to a lesser extent compared to the FUS effect, and the cytoplasmic expression of hnRNP A1 may be linked to ALS and FTD by promoting protein aggregation [134]. Interestingly, the expression levels of hnRNP A1 and its regulatory factor miR-590-3p have been reported to be altered in blood cells from patients with FTLD [135]. One of the important functions of hnRNP H is splicing of the target transcripts, and the insolubility of hnRNP H correlates with that of the ALS-linked RBPs, such as TDP-43 and FUS [136]. hnRNP H directly binds to and suppresses a specific long noncoding RNA by functioning as a sponge of miR-663a [137]. The expression of miR-663a has been reported to be significantly increased in the blood of sporadic ALS patients [138]. RALY, the RBP associated with lethal yellow mutation, is a member of the hnRNP family. RALY regulates the arginine methylation of FUS catalyzed by Protein arginine N-methyltransferase 1 (PRMT1), and thereby, its dysregulation leads to the FUS aggregation involved in the etiology of ALS [139]. RALY acts as a key regulatory component in the Drosha complex and promotes the post-transcriptional processing of a specific subset of miRNAs (miR-483, -676 and -877), which, in turn, regulates the genes related to mitochondrial metabolism, as well as cell proliferation and apoptosis, under oxidative stress [140]. RBM17 and RBM45 play important roles in extensive cryptic splicing and chromatin recruitment, respectively, and the alteration of these proteins has been implicated in ALS and FTD [141,142]. A computational analysis revealed that miR-150-5p functions together with RBM17 in the expression of Vascular endothelial growth factor (VEGF) [143]. VEGF is a proangiogenic factor that confers neuroprotection by promoting neuron survival, which is reduced in an ALS mouse model [144], and miR-150-5p was identified as significantly downregulated in sporadic ALS [145]. HuD has been reported to regulate ALS-associated SOD1 expression during oxidative stress in differentiated neuroblastoma cells and the sporadic ALS motor cortex [146]. HuD binds to and regulates circular RNAs derived from genes regulating neuronal development and synaptic plasticity and may act as a sponge for the miRNAs such as miR-125a-5p, which has previously been reported to be associated with aging [147,148]. Matrin-3 was recently found to be strongly linked to the pathogenesis of ALS [149,150,151] and has been shown to regulate processing of the synaptic miR-138-5p [152] (Table 2).

### 4.2. Alzheimer’s Disease

AD is an irreversible age-related ND characterized by progressive dementia developed in middle or later life [153]. The pathological hallmarks are depositions of Amyloid β (Aβ) plaques and neurofibrillary tangles composed of abnormally phosphorylated tau protein in the brain. Interestingly, the recent lines of evidence indicate that brain insulin resistance and insulin deficiency are associated with cognitive impairment and neurodegeneration, particularly those in AD [154].

In regard to the associations of RBPs and miRNAs with AD pathology, oligomeric A stimulation has been shown to induce a decrease of RBPs, including hnRNP K, and subsequent neurodegeneration in AD pathogenesis [155]. Intronic miR-7 is co-transcribed along with the host gene hnRNP K, which was recently shown to be involved in the actions of insulin and to regulate insulin signaling and A levels in the brain during the progression of AD [156]. Further, Msi proteins were found to be present in an oligomeric state in the brains of patients with AD, and aberrant interactions between Msi and tau in the nuclei appear to contribute to the pathogenesis of AD [157,158]. Msi2 has been reported to regulate the biogenesis of brain-enriched miR-7, a key element of the insulin signaling pathway and AD [107]. RBM8A is significantly downregulated in AD. RBM8A might regulate the components of key complexes in the autophagy pathway, and its dysregulation might contribute to the pathogenesis of AD [159]. RBM8A and miR-29a have been shown to competitively regulate the proliferation and differentiation of retinal progenitors [160]. miR-29a has been reported to be decreased in the brain and blood of AD patients and to be involved in A generation [161]. RBM45 forms nuclear and cytoplasmic inclusions in the neurons and glia in AD, as well as ALS and FTLD-TDP, through a persistent association with nuclear stress bodies, thereby promoting its own aggregation into insoluble inclusions [162]. RBM45 acts as a component of the spliceosome, and its splicing process is regulated by miR-4454 in response to insulin signaling in a cell type-dependent manner, although such regulation has been reported only in the prostate glands so far [163]. The direct interaction of tau with TIA1 in the presence of RNA promotes the phase separation of tau and the formation of toxic oligomeric tau [164]. TIA1 plays an important role in the translational silencing of Methyl CpG-binding protein 2 (MECP2), which is important for normal brain development and physiology, in combination with Pummilio1, which recruits miR-200a and -302c, while HuC, instead of TIA1, predominates in neurons, resulting in a switch to translational enhancement [165]. miR-200a may be involved in AD pathology, although whether it contributes to neurotoxicity or neuroprotection is still controversial [166,167]. CELF1, which regulates the networks of postnatal alternative splicing transitions in the nucleus and mRNA stability and translation in the cytoplasm, and the misregulation and mislocalization of CELF1 have been implicated in the tau-mediated mechanisms in AD [168]. The function of CELF1 on the 3′-UTR of specific genes is indicated to be antagonized by miR-574-5p and miR-206 [169,170]. Both miR-574-5p and miR-206 were found to be dysregulated in AD patients [171,172]. In addition, CELF1 jointly acts as a modulator of translation with miR-222, a significantly lowered miRNA in the blood of AD patients [173,174], and enhances the deadenylation and degradation of miR-122, an upregulated miRNA in the brain with AD patients [175,176]. FMRP and hnRNP C oppositely regulate Amyloid precursor protein (APP) translation in the hippocampus of an AD animal model, and their dysregulation was observed in the hippocampus synaptosomes of patients with sporadic AD [177]. The FMRP-dependent miR-128-3p regulation of metabolic Glutamate receptor 5 (mGluR5) expression in astroglia is important for neuronal development, and its upregulation has been correlated with impaired Aβ degradation in monocytes from patients with sporadic AD [178,179]. On the other hand, the FMRP-mediated axonal delivery of miR-181d negatively regulates axon elongation by locally targeting the transcripts of Microtubule-associated protein 1B (MAP1B) and Calmodulin 1 (Calm1) in the primary sensory neurons [180]. Several lines of evidence show that miR-181 family members, including miR-181d, are involved in AD pathogenesis, although their roles in AD are still controversial, because different research groups have reported different directions of regulation on miR-181 expression in AD patients and models [181]. FMRP-associated miR-125b and miR-132 have opposite effects on dendritic spine morphology and synaptic physiology in hippocampal neurons through the regulation of NMDA receptor signaling, possibly mediated by Argonaute proteins [182]. miR-125b is one of the AD biomarker miRNAs [182,183], and miR-132 is known as a member of the neurotoxic miRNA family in AD pathogenesis [184]. hnRNP C was identified as a main splicing factor, and it was shown to be essential for the generation of circular RNA-sorting nexin 5 (circSnx5), which acts as an miR-544 sponge to attenuate miRNA-mediated target depression [185]. Recently, a polymorphism of miR-544 was identified and shown to be associated with the risk of late-onset AD (LOAD) [186]. RCAN1 plays an important role in the pathogenesis of Down syndrome, as well as AD, through inhibition of the NFAT and NF-κB signaling pathways and thereby induces neuronal apoptosis [187]. RCAN1 mRNA has been reported to be regulated by miR-324-5p, miR-4738-3p and lncRNA miR-497-HG [188], although a PubMed search suggests that there is no evidence of an interaction between these miRNAs and AD pathogenesis. Insoluble aggregates composed of tau with RBPs, including EWS and PABP, are known markers for stress granules, which are observed in the tauopathies represented by AD [189]. EWS controls the levels of the miRNAs, such as miR-29b and miR-18b, via Drosha, leading to autophagy dysfunction and impaired cellular development [190]. In addition, EWS is involved in the post-transcriptional gene regulation via miR-125a and miR-351 that results in the deregulation of autophagy inhibition [191]. Both miR-29b and miR-18b are known to be AD biomarkers [192], and the expressions of miR-125a and miR-351 are altered in an AD model [193,194]. PABP has a role in promoting the association of miRISC with mRNAs regulated by miR-2, which targets neural genes according to a computational analysis [114]. AD brain homogenates induce the aggregation of soluble ribosomal protein U1-70K to make U1-70K detergent-insoluble, and its specific domains interact with pathological Tau specifically from the AD brain [195]. Although there appear to be no reports on the association of U1-70K with miRNAs, based on a PubMed search, it would be interesting to find a link between miRNAs and ribosomal proteins. Finally, TDP-43 also plays an important role in AD pathologies—namely, TDP-43 forms oligomeric assemblies associated with tau in cytoplasmic stress granules [196]. TDP-43 could modify the levels of miRNAs through direct interactions with let-7 and pre-miR-663, which are involved in the pathogenesis of AD and Down syndrome [197]. As described above, TDP-43 also affects the expression of miR-181c-5p and miR-27b-3p through Drosha and/or Dicer processing and vice versa [124]. A lower expression of miR-181c-5p in the serum is associated with increased levels of plasma A [198], and a reduced expression of miR-27a-3p is observed in the cerebrospinal fluid of patients with AD [199] (Table 3).

### 4.3. Parkinson’s Disease

PD is the second-most common ND after AD and is clinically characterized by resting tremors, rigidity, akinesia and postural instability [200]. PD is characterized as a progressive, late-onset movement disorder that is affected by dopaminergic neurodegeneration in the substantia nigra (SN). Lewy bodies, which are eosinophilic neuronal inclusions that contain both α-synuclein and ubiquitin, are pathological hallmarks of PD [201].

The RNA-binding activity of the recessive parkinsonism protein DJ-1 supports involvement in multiple cellular pathways [202]. DJ-1 is capable of binding a series of target mRNA molecules that is related to antioxidant defenses and plays a role in coordinating the responses to oxidative damage [203]. DJ-1 modulates miR-221 to promote neuronal survival against oxidative stress [204]. miR-221 was significantly decreased in PD patients, and serum miR-221 was positively correlated with the United Parkinson’s disease rating scale score in PD patients [205]. In addition, DJ-1 efficiently regulates inflammation by maintaining the expression of Suppressor of cytokine signaling 1 (SOCS1), a well-characterized negative feedback regulator of cytokine-signaling molecules, through the regulation of miR-155 [206]. miR-155 is often dysregulated in patients with any of several NDs, including PD, as well as in the animal models of these NDs, and plays an important role in -synuclein-induced inflammatory responses [13,207]. Moreover, the analysis of SNP haplotypes in combination with the PD risk has revealed that HuD is significantly associated with the age-at-onset trait in PD [208]. HuD competes with miR-30a-3p for binding to the 3’-UTR of p27 mRNA in pancreatic neuroendocrine tumors [209], and miR-30a-3p has been reported to be upregulated in the serum of PD patients [210]. In addition, TTP protects against dopaminergic oxidative damage by destabilizing NADPH oxidase 2 mRNA in the PD model [211] and has been identified as a post-transcriptional repressor of the genes specific for the nervous system that is counteracted by miR-9 [212]. The dysregulation of miR-9 has been reported in PD, and the dysregulation of miR-9 has been shown to exert a neurotoxic role in a PD model [210,213] (Table 4).

### 4.4. Huntington’s Disease

HD is caused by the expansion of CAG trinucleotide repeats (in excess of 38 repeats) on chromosome 4 in exon 1 of the gene coding huntingtin (HTT) with autosomal-dominant inheritance [214]. HD patients show hyperkinetic movement disorders due to basal ganglion dysfunction. Since HD is an inherited disease, there are fewer clinical studies profiling the miRNAs involved in HD than profiling the miRNAs involved in ALS and AD. However, several studies suggest that the disease progression of HD might be related to the dysfunction of miRNAs. In addition, it has been implicated that AD often occurs together with HD, although further studying is needed to determine the cooccurrence of AD and HD because of the small sample size [215], so that the features on the dysfunction of RBPs and miRNAs may overlap between the two NDs.

The pathogenically decreased expression of Splicing regulatory glutamic acid and lysine rich protein 1 (SREK1), which is an interacted protein of SRSF6, leads to a TAF1 deficit in HD pathogenesis [216]. SRSF6 mediates alternative splicing, and SRSF6 is targeted by miR-193a-5p [217]. miR-193a-5p has been identified as a significantly altered miRNA in autism spectrum disorder, indicating its possible relation to neural development [218]. One of the long noncoding RNAs, Tcl1 upstream neuron-associated (TUNA), has been shown to be associated with HD pathogenesis, the function of which is to mediate the recruitment of RBPs such as PTBP1, hnRNP K and nucleolin to the promoter of the genes for pluripotency and neural stem cell marker [219]. PTBP1 plays an important role in the blood–brain tumor barrier by interacting with circular RNA_001160 and regulating the function of miR-186-5p and miR-195-5p [220,221]. It has been reported that miR-186-5p could suppress β-site amyloid precursor protein cleaving enzyme 1 (BACE1) expression by directly targeting its 3′-UTR and was observed to be decreased in the brain of aged mice. miR-195-5p has been reported to be reduced in the brains of patients with AD, along with disease progression, and indicated to be involved in the pathogenesis of AD by targeting APP and BACE1 [222,223,224]. hnRNP K promotes neuronal apoptosis through the miR-107-mediated activation of long noncoding RNA rhabdomyosarcoma 2 induced by oxygen–glucose deprivation [225], and this miRNA is identified as an AD-related factor that is decreased in the brains of patients with AD and may accelerate disease progression [226].

HuR stabilizes HTT mRNA by interacting with its exon 11 in a mutant HTT-dependent manner, which might contribute to HD progression through the enhancement of mutant HTT protein accumulation [227]. HuR is known to be involved in brain-enriched miR-7 biogenesis, together with Msi2, as described in AD pathogenesis (see Section 4.2) [107]. SAFB1 is involved in the processing of coding and noncoding RNAs, splicing and dendritic functions, and the expression of this protein is altered in the post-mortem brain tissue of HD patients [228]. SAFB1 has been reported to play an important role in the processing of miRNAs such as HD-associated miR-19a from the miR-17-92 cluster and neuronal function [229,230] (Table 5).

### 4.5. Multiple System Atrophy (MSA)

MSA is a devastating ND representing parkinsonism, cerebellar ataxia, autonomic dysfunction and pyramidal signs [233]. MSA that is clinically predominated by parkinsonism is defined as MSA-P, while that predominated by cerebellar ataxia is called MSA-C [231]. MSA patients show significant brain atrophy of the putamen, cerebellum, pons or middle cerebellar peduncle, with mild cortical atrophy in the frontal lobes. The mean age at the onset of the symptoms is around 60 years, and the mean survival from the onset is around 10 years. At present, there is no disease-modifying therapy, and only symptomatic therapies, such as levodopa, are available for clinical use.

It has been reported that miR-96-5p is upregulated as a disease-specific miRNA that is dysfunctional in either MSA patients or an animal model of MSA, and Excitatory amino acid carrier 1 (EAAC1) has been found to be downregulated in MSA model mice [232]. Recently, miR-96-5p has been reported to indirectly regulate EAAC1 expression through NOVA1 by inhibiting an endogenous EAAC1 regulator, glutamate transport-associated protein 3-18 (GTRAP3-18), the deletion of which promotes neuroprotection in the mouse hippocampus [231,234] (Table 5).

## 5. Conclusions

The presence of insoluble inclusions in and around neurons is a common pathological feature of most NDs [1]. An analysis of these inclusions revealed that they contain several misfolded and aggregated proteins, including several RBPs. Under normal conditions, RBPs play important roles in splicing, stability, transcription and translation, contributing to the precise regulation of gene expression [3]. However, genetic mutations and/or the dysregulation of several RBPs under cellular stress conditions contribute to pathogenic assembly, which has been identified as a cause of NDs [7].

RBPs also have a crucial function in miRNA biogenesis and metabolism, and miRNAs are precisely and accurately regulated by several RBPs [102]. Since the evidence gathered to date indicates that the abnormal expression and function of miRNAs is correlated with ND pathogenesis and progression, the aberrant expression and/or misregulation of RBPs may be one of the primary factors for miRNA dysfunction [235]. Since the growing lines of evidence indicate that miRNAs and RBPs interact and cooperate in the complex process of regulating the gene expression, understanding the mechanisms underlying the interplay of miRNAs and RBPs could form a basis for the development of ND therapeutics.

Although many dysregulated miRNAs and RBPs have been observed in patients with NDs, it is still unclear whether their aberrant expression is a cause, a consequence or a compensatory mechanism of the respective diseases. In any case, it is clear that miRNAs could function as either a therapeutic agent or biomarker by permitting the regulation of the genes that play key roles in ND onset or progression. Further investigation will be needed to elucidate the relationship between the NDs and the dysfunction of the miRNAs and RBPs for the development of such novel therapeutic agents.

## Figures and Tables

**Figure 1 ijms-22-05292-f001:**
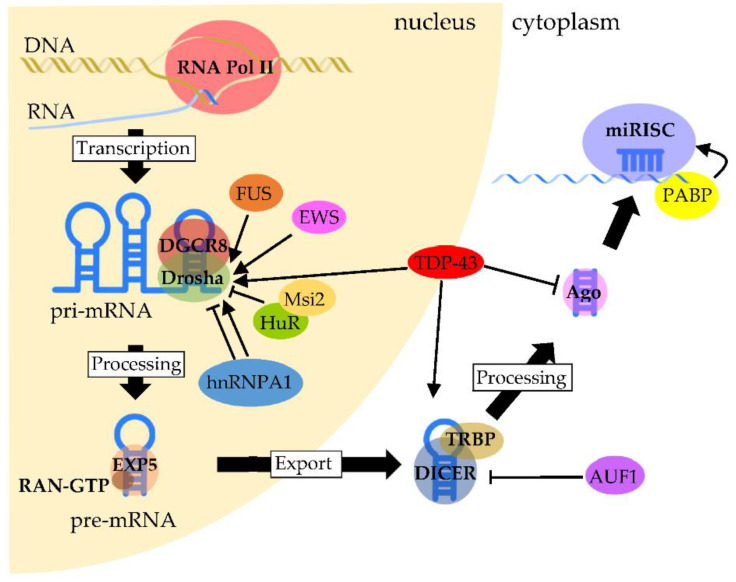
Regulation of RBPs in miRNA biogenesis and processing. Pri-miRNAs are transcribed mainly by RNA polymerase II. The processing of pri-miRNA is mediated by a complex formed between Drosha and DGCR8, called Microprocessor, to generate pre-miRNA. TDP-43 facilitates the association of the Drosha complex with pri-miRNAs, whereas FUS and EWS promote Drosha recruitment to chromatin. HuR-mediated Msi2 binding to pri-miRNA inhibits Drosha cleavage. The function of hnRNP A1 is complexed, since it contributes to either the inhibition or stimulation of processing, which depends on the pri-miRNA. After pre-miRNAs are exported through the binding of Exportin-5 with RAN-GTP, pre-miRNAs are processed by Dicer. The processing of pre-miRNAs is facilitated by TDP-43 through the binding of Dicer with pre-miRNAs, whereas Dicer expression is negatively regulated by AUF1. On the other hand, TDP-43 negatively regulates the formation of miRISC from the Ago protein. PABP binds to the poly(A) region of target mRNA and promotes miRISC recruitment.

**Table 1 ijms-22-05292-t001:** List of domains and estimated binding motifs of RBPs based on the protein database.

RBP	Domain [22]	Binding Motif [23,24]	
RBM	more than one RRM ZF domain (RBM4, 4B, 5, 10, 20, 22, 26, 27)	GCGC(G)	RBM4, 4B, 8A
		GAAGG	RBM5
		AUCCA	RBM6
		GUGUG	RBM24, 38
		GUAG	RBM28
		UACA/UU	RBM41
		AACUA	RBM42
		GACGAC	RBM45
		AUC/GA	RBM46, 47
		AUAUA	RBMS1, S2, S3
		CCA/C	RBMX
		CU/ACAA	RBMXL1, XL2, XL3, Y1A1, Y1B, Y1D, Y1E, Y1F, Y1J
hnRNP	one to three RRMs (except hnRNP E, F, H, K, U) three qRRMs (hnRNP F, H)three KH domains (hnRNP E, K)acidic-rich domain (hnRNP C, Q, R, U)	UAGGG	hnRNP A1, A1L2, A2B1, A3
		UUUUU	hnRNP C, CL1
		GGGAGG	hnRNP F, H1, H2
		CCAA/UC/ACC	hnRNP K
		ACAC	hnRNP L
		ACANACA	hnRNP LL
		UAAU/AU/A	hnRNP DL
TDP-43	two RRMs, G-rich region	GAAUG	
FET proteins	RRM, three RGG boxes, SYGQ-rich region, ZF domain	poly(G)	
ELAV/Hu family	three RRMs, hinge region	poly(U)	
CELF	three RRMs, hinge region	GUGU	
RCAN	RRM, LxxP motif, PxIxIT-like motif, TxxP motif	n.d.	
Msi	two RRMs	UAG	
FMRP	two KH domains, RGG box	GAC	
TIA	three RRMS, Q-rich domain	poly(U)	
TTP	two ZF domains,	AU-rich	
SRSF	one or two RRMs, RS domain	GGAGGA/G	SRSF 1, 2, 4, 6, 9
		(A)GCAG(C)	SRSF 2, 4, 5, 8, 10
		GA/GUCAACCNGGCGACA/CG	SRSF 4, 6
NOVA	three KH domains	U/CCAU/C	
Matrin3	two RRMs, two ZF domains	AUCUU	
PABP	one or four RRMs	poly(A)	
SAFB	SAP domain, RRM, RG/RGG domain	AG-rich	

**Table 2 ijms-22-05292-t002:** List of interactions between the RBPs and miRNAs in the ALS and the FTD.

Associated RBPs	Related miRNAs	Target Gene(s)	Relation between RBPs and miRNAs	miRNA Abnormalities in Diseases
TDP-43	miR-181c-5p	TDP-43	forms a negative feedback loop in miRNA processing [124]	is reduced in the spinal cord of ALS patients [125]
	miR-27b-3p			
	miR-8485	NRXN1	forms an miRISC complex to decrease genes related to synaptic formation [126]	n.d.
FUS/TDP-43	miR-183/96/182	PP1	regulates miRNA biogenesis, which is related to memory formation [129]	is reduced in the frontal cortex of ALS patients [129]
TAF15	miR-17-92	CDKN1A/p21	regulates miRNA biogenesis to decrease genes related to cell proliferation [132]	is reduced in an SOD1^G93A^ mouse model [133]
hnRNP A1	miR-590-3p	n.d.	is cooperatively regulated in neuronal maintenance ? [135]	is increased in the blood cells of FTD patients [135]
hnRNP H	miR-663a	CDK4, CDK6	is cooperatively regulated in cellular senescence by targeting genes related to the cell cycle [137]	is increased in the blood cells of ALS patients [138]
RALY	miR-483	ATP5I, ATP5G1, ATP5G3, CYC1	regulates miRNA biogenesis to decrease genes related to mitochondrial metabolism [140]	n.d.
	miR-676			
	miR-877			
RBM17	miR-150-5p	VEGF-A	is cooperatively regulated in genes related to growth factors [143]	is reduced in the CSF of ALS patients [145]
HuD	miR-125a-5p	CAMSAP3	cooperatively regulates circRNA in aging by targeting genes related to neuronal polarization [148]	is reduced in aging mice [147]
Matrin-3	miR-138-5p	n.d.	inhibits miRNA biogenesis in the regulation of neuronal development [152]	n.d.

n.d.: not detected.

**Table 3 ijms-22-05292-t003:** List of interactions between RBPs and miRNAs in an AD.

Associated RBPs	Related miRNAs	Target Gene(s)	Relation between RBPs and miRNAs	miRNA Abnormalities in Diseases
hnRNP K	miR-7	ISNR, IRS-2, IDE	intronic miRNA and its host genes, regulating genes involved in the insulin signaling in neuronal cells [156]	is elevated in the brain of AD patients [156]
Msi2	miR-7	n.d.	regulates miRNA biogenesis, which is related to neuronal differentiation [107]	is elevated in the brain of AD patients [156]
RBM8A	miR-29a	RBM8A	competitively regulates cell proliferation and differentiation [160]	is decreased in the brain and blood cells of AD patients [161]
RBM45	miR-4454	INSR, GLUT4	competitively regulates insulin response [163]	n.d.
TIA1/HuC	miR-200a	MECP2	promotes or inhibits the action for miRNA recruitment on genes critical for neuronal maturation [165]	is altered in an AD model and in AD patients [166,167]
	miR-302c			n.d.
CELF1	miR-574-5p	mPGES-1	competitively regulates genes related to inflammation and proliferation [169]	is increased in the cortex of APP KO mice [171]
	miR-206	MyoD	competitively regulates genes related to differentiation [170]	is increased in the brain of an AD mouse model [172]
	miR-222	CDK4	cooperatively regulates cellular senescence by targeting genes related to the cell cycle [174]	is decreased in the serum of AD patients [173]
	miR-122	BCKDK, ALDOA, NDRG3, CCNG1, CAT1	enhances destabilization of miRNA in the regulation of energy metabolism, stress response and cell cycle [176]	is increased in the brain of AD patients [175]
FMRP	miR-128-3p	mGluR5	competitively regulates astroglial development by targeting glutamate receptor genes [179]	is increased in monocytes of AD patients [178]
	miR-181d	MAP1B, Calm1	cooperatively regulates axon elongation by targeting genes crucial for calcium signaling [180]	is altered in an AD model and in AD patients [181]
	miR-125b	NR2A	cooperatively regulates synaptic strength by targeting an NMDA receptor subunit gene [182]	is increased in the CSF of AD patients [183]
	miR-132	p250GAP?	cooperatively regulates synaptic strength possibly by targeting genes related to Rho family GTPase [182]	is decreased in the exosome of AD patients [184]
hnRNP C	miR-544	SOCS1	competitively regulates circRNA in inflammation by targeting genes related to cytokine signaling [185]	is increased in high LOAD risk SNPs [186]
RCAN1	miR-324-5p	n.d.	inhibits RBP expression [188]	n.d.
	miR-4738-3p			
EWS	miR-29b	Col4a1, CTGF	negatively regulates miRNA biogenesis in cell senescence by targeting genes related to differentiation [190]	is decreased in the serum of AD patients [192]
	miR-18b			is increased in serum of AD patients [192]
	miR-351	UVRAG	negatively regulates miRNA biogenesis in autophagy by targeting genes related to the autolysosomal pathway [190]	is increased in the hippocampus of an AD mouse model [193]
	miR-125a			is increased in AD cellular model [194]
PABP	miR-2	n.d.	facilitates miRISC onto the 3’-UTR of targeting gene [114]	n.d.

n.d.: not detected.

**Table 4 ijms-22-05292-t004:** List of interactions between the RBPs and miRNAs in PD.

Associated RBPs	Related miRNAs	Target Gene(s)	Relation between RBPs and miRNAs	miRNA Abnormalities in Diseases
DJ-1	miR-221	n.d.	positively regulates miRNA expression in neurite outgrowth and neuronal differentiation [204]	is decreased in serum of PD patients [205]
	miR-155	SOCS1	negatively regulates miRNA expression in inflammation by targeting genes related to cytokine signaling [206]	is increased in SNpc of a PD mouse model [207]
HuD	miR-30a-3p	p27	competitively regulates proliferation by targeting gene related to cell cycle [209]	is increased in the serum of PD patients [210]
TTP	miR-9	Tubb3, HuB, HuC, HuR, NOVA1	inhibits RBP regulating neuronal differentiation and targeting neuronal markers [212]	is increased in PD patients and a mouse model of PD [210,213]

n.d.: not detected.

**Table 5 ijms-22-05292-t005:** List of interactions between the RBPs and miRNAs in HD and MSA.

Disease	Associated RBPs	Related miRNAs	Target Gene(s)	Relation between RBPs and miRNAs	miRNA Abnormalities in Diseases
HD	SRSF6	miR-193a-5p	OGDHL, ECM1	forms a negative feedback loop in cancer cell migration and invasion by regulating alternative splicing [217]	is decreased in the blood of Autism patients [218]
	PTBP1	miR-195-5p	ETV1	competitively regulates circRNA in BTB permeability by targeting genes related to tight junction [220]	is decreased in the brain and CSF of AD patients [222]
		miR-186-5p	Occludin	competitively regulates circRNA in BTB permeability by targeting genes related to tight junction [221]	is decreased in the brain of aged mouse [224]
	hnRNP K	miR-107	Bcl2l2	promotes miRNA activation via circRNA in neuronal apoptosis by targeting apoptotic regulating genes [225]	is decreased in the brain of AD patients [226]
	HuR	miR-7	n.d.	regulates miRNA biogenesis, which is related to neuronal differentiation [107]	is elevated in the brain of AD patient [156]
	SAFB1	miR-19a	n.d.	regulates miRNA biogenesis [229]	is decreased in a cellular model of HD [230]
MSA	NOVA1	miR-96-5p	GTRAP3-18	inhibits RBPs regulating neuroprotection and targeting neurotoxic genes [231]	is decreased in the frontal cortex of MSA patients [232]

n.d.: not detected.

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
