# Peer review of "Interplay of RNA-Binding Proteins and microRNAs in Neurodegenerative Diseases"

_ijms, 2021, doi:10.3390/ijms22105292_

Round 1

Reviewer 1 Report

Manuscript by Kinoshita et al. provides a valuable summary of disrupted interactions (direct or indirect) between proteins that bind RNAs and miRNAs, in case of various neurodegenerative diseases. The authors quite comprehensively present information about RNA-binding proteins, involvement of these proteins in miRNA biogenesis and what is known about connections in RBP and miRNA disruptions in several diseases resulting in neurodegeneration. My general opinion is that it would be valuable to include more thoughts about analysis of the data, as the authors mainly are more enumerating and describing the observations made, without discussing them. In some sections the authors need to be more precise.

I have several detailed suggestions for the authors:

  • I find chapter 3 as redundant, as information placed there could be included in chapter 5.
  • The authors should be careful not to repeat some information throughout the text.
  • Abstract, line 18-19, I find it misleading to use “Recently” in this sentence, as RBP are known for a long time to be involved in miRNA biogenesis regulation. Of course another protein factors are being revealed, but it could be worded differently. Actually, references 109-119 which refer to this issue are not very new.
  • In chapter 4 it would be worth to include some references in the introduction about miRNAs, e.g. for their mechanism of action (like doi: 10.1007/s00018-013-1551-6)
  • Line 375: it should be: “Most miRNA genes are located…)
  • Line 382: Microprocessor should be written in capital letter
  • Figure 1, figure legend, the last sentence is unfinished
  • Line 443: the sentence about what caused NDs is a bit confusing. The direct cause for some of these diseases are genetic mutations which further result in described dysregulations.
  • Chapter 5.4: line 657, the conclusion from reference 218 is not correct. The authors of original paper conclude that “Further study is needed to determine if there is a higher incidence of AD in persons with HD compared to the general population.”
  • Line 669: placing information about AD here is not clear
  • Line 675: information about HuR should start from another line.
  • Definitely Conclusions could be supported by more discussion, maybe referring to disruptions of (e.g. recently reviewed in doi.org/10.3390/ijms21249582)

Author Response

We are grateful to you for the insightful comments and useful suggestions that have helped us to improve our paper.  As indicated in the responses that follow, we have taken all these comments and suggestions into account in the revised version of our paper.  I have several detailed suggestions for the authors:

Comment 1: I find chapter 3 as redundant, as information placed there could be included in chapter 5. The authors should be careful not to repeat some information throughout the text.

Response: In accord with this suggestion, we removed chapter 3 and some of the information moved to section 1 and 4 (line 64-77, 588).

Comment 2: Abstract, line 18-19, I find it misleading to use “Recently” in this sentence, as RBP are known for a long time to be involved in miRNA biogenesis regulation. Of course another protein factors are being revealed, but it could be worded differently. Actually, references 109-119 which refer to this issue are not very new.

Response: I accord with this suggestion, we removed the term “Recently” and added “In addition” instead (line 18).

Comment 3: In chapter 4 it would be worth to include some references in the introduction about miRNAs, e.g. for their mechanism of action (like doi: 10.1007/s00018-013-1551-6)

Response: In accord with this suggestion, we added suggested article as a reference (line 351).

Comment 4: Line 375: it should be: “Most miRNA genes are located…)

Response: In accord with this suggestion, we added “genes” after “Most miRNA” in this sentence (line 347).

Comment 5: Line 382: Microprocessor should be written in capital letter

Response: In accord with this suggestion, we capitalize the first letter of “Microprocessor” (line 354, 355-356).

Comment 6: Figure 1, figure legend, the last sentence is unfinished

Response: Thank you for pointing this out. We have added the rest of the sentence (line 542).

Comment 7: Line 443: the sentence about what caused NDs is a bit confusing. The direct cause for some of these diseases are genetic mutations which further result in described dysregulations.

Response: In accord with this suggestion, we added following sentences; “Although the genetic mutations which cause the pathogenesis of familial NDs have been well studied, the majority of NDs are sporadic with the exception of HD. Our knowledge of sporadic NDs is unfortunately not sufficient to understand their molecular etiologies even though their pathologies are well established.  Several lines of evidence indicate that onset of sporadic NDs is due to the synergistic effect of genetic polymorphism, environmental factors or their combination with overlapping disease mechanisms, as seen in the familial disease pathologies. In fact, gene expression profiling from clinical studies has revealed a profound dysregulation of gene expression in the tissues of patients with NDs, suggesting that post-transcriptional and/or post-translational regulation of gene expression would clearly affect the onset or progression of NDs.” (line 551-560)

Comment 8: Chapter 5.4: line 657, the conclusion from reference 218 is not correct. The authors of original paper conclude that “Further study is needed to determine if there is a higher incidence of AD in persons with HD compared to the general population.”

Response: In accord with this suggestion, we changed the sentence as follows; “it has been implicated that AD often occurs together with HD although further study is needed to determine the co-occurrense of AD and HD because of small sample size”. (line 801-803)

Comment 9: Line 669: placing information about AD here is not clear

Response: In accord with this suggestion, we changed the sentences as follows: It has been reported that miR-186-5p could suppress β-site amyloid precursor protein cleaving enzyme 1 (BACE1) expression by directly targeting its 3’-UTR and was observed to be decreased in the brain of aged mouse. miR-195-5p has also been reported to be reduced in the brains of patients with AD along with disease progression, and indicated to be involved in pathogenesis of AD by targeting APP and BACE1. (line 819-824)

Comment 10: Line 675: information about HuR should start from another line.

Response: In accord with this suggestion, we inserted line feeds before HuR sentences. (line 829)

Comment 11: Definitely Conclusions could be supported by more discussion, maybe referring to disruptions of (e.g. recently reviewed in doi.org/10.3390/ijms21249582)

Response: In accord with this suggestion, we added several reference including your suggested one and following sentences: “Although many dysregulated miRNAs and RBPs have been observed in patients with NDs, it is still unclear whether their aberrant expression is a cause, a consequence or a compensatory mechanism of the respective diseases. In any case, it is clear that miRNA could function as either a therapeutic agent or biomarker, by permitting the regulation of genes that play key roles in the ND onset or progression. Further investigation will be needed to elucidate the relationship between NDs and dysfunction of miRNAs and RBPs for the development of such novel therapeutic agents.” (line 879-885)

Reviewer 2 Report

Review of a manuscript “Interplay of RNA-binding proteins and microRNAs in neurodegenerative diseases by Kinoshita and coauthors” submitted to IJMS.

Neurodegenerative diseases are severe human diseases associated with the death of neurons in specific brain arras. These disorders are usually accompanied with the formation of inclusions and deposits consisting of misfolded and aggregated proteins. These inclusions contain RNA-binding proteins among other components. There is no therapy affecting the main course of these disorders, so it is important to conduct basic research to find the cure and to develop diagnostic tools for early diagnostic. The authors discuss recent results on a role of RNA-binding proteins and microRNAs in neurodegeneration. This is important direction of biomedical research and the data and opinions presented in the manuscript will be interesting for the readership of the IJMS.

The following corrections should be made:

Abstract:

Line 12: “The number of patients with neurodegenerative diseases (NDs) is increasing along with the increasing number of older adults, and threatens to create a medical and social crisis”.

This is an awkward sentence, which can be corrected as follows:” The number of patients with neurodegenerative diseases (NDs) is increasing along with the growing number of older adults. This escalation threatens to create a medical and social crisis.”

Lines 20-21: “Since dysfunction of both RBPs and miRNAs is often observed in several NDs, the interplay and cooperation among RBPs and miRNAs in brain function would be important for understanding NDs and the development of effective therapeutics.”

This is a long and blurred sentence which can be split for easier reading: ”Dysfunction of both RBPs and miRNAs is often observed in several NDs. Thus, the data about the interplay among RBPs and miRNAs and their cooperation in brain functions would be important to know for better understanding NDs and the development of effective therapeutics.”

Introduction:

Line 60: “Recently, the function of miRNAs has been clarified to be precisely organized by a large

number of RBPs.” The sense of this sentence is unclear. Should be rewritten in a more clear way.

15 Poly(A)-binding protein (PABP)

 Line 268:”…and was shown to stimulate the translation of mRNAs harboring a poly(A) tail in principle [83]. “

It is unclear what the authors want to say by “in principle”. Should be clarified or deleted.

Lines 306-316 The manuscript is sometimes overflowed by well-known basic information. For example, the fragment beginning at line 306 “The ribosome is a complex cellular machine which is responsible for protein synthesis … and ending at line 316 “…a new nomenclature has been proposed to resolve this problem  [94] contains general information. The authors should make the manuscript more succinct and concise.

Table 2. List of interactions between RBPs and miRNAs in NDs

The Table is overloaded with data and hard to read. It should be either split or truncated.

5.3 Parkinson’s disease

Lines 624-625: “Lewy bodies, which are eosinophilic neuronal inclusions that contain both α-synuclein and ubiquitin, are pathological hallmarks of PD.”

The authors should add here a reference on a recent review: Emamzadeh  et al. “Parkinson’s disease: Biomarkers, Treatment, and Risk Factors. Front. Neurosci., 018, 12:612 | https://doi.org/10.3389/fnins.2018.00612.

  1. Conclusion

Line 714 “…aberrant expression and/or misregulation of RBPs may one of the primary factors for miRNAs dysfunction”. This is incomplete sentence. Presumably, the authors want to say :” …aberrant expression and/or misregulation of RBPs may be one of the primary factors for miRNAs dysfunction”.

Author Response

We are grateful to you for the insightful comments and useful suggestions that have helped us to improve our paper.  As indicated in the responses that follow, we have taken all these comments and suggestions into account in the revised version of our paper.  I have several detailed suggestions for the authors:

Comment 1: Line 12: “The number of patients with neurodegenerative diseases (NDs) is increasing along with the increasing number of older adults, and threatens to create a medical and social crisis”.This is an awkward sentence, which can be corrected as follows:” The number of patients with neurodegenerative diseases (NDs) is increasing along with the growing number of older adults. This escalation threatens to create a medical and social crisis.”

Response: We changed the sentence as you suggested. (line 12)

Comment 2: Lines 20-21: “Since dysfunction of both RBPs and miRNAs is often observed in several NDs, the interplay and cooperation among RBPs and miRNAs in brain function would be important for understanding NDs and the development of effective therapeutics.” This is a long and blurred sentence which can be split for easier reading: ”Dysfunction of both RBPs and miRNAs is often observed in several NDs. Thus, the data about the interplay among RBPs and miRNAs and their cooperation in brain functions would be important to know for better understanding NDs and the development of effective therapeutics.”

Response: We changed the sentence as you suggested. (line 19-21)

Comment 3: Line 60: “Recently, the function of miRNAs has been clarified to be precisely organized by a large number of RBPs.” The sense of this sentence is unclear. Should be rewritten in a more clear way.

Response: In accord with this suggestion, we added following sentence; as RBPs control different stages of miRNA biogenesis and their localization, degradation and activity. (line 83-84)

Comment 4:  Line 268:”…and was shown to stimulate the translation of mRNAs harboring a poly(A) tail in principle [83]. “It is unclear what the authors want to say by “in principle”. Should be clarified or deleted.

Response: In accord with this suggestion, we deleted the word “in principle”. (line 291)

Comment 5: Lines 306-316 The manuscript is sometimes overflowed by well-known basic information. For example, the fragment beginning at line 306 “The ribosome is a complex cellular machine which is responsible for protein synthesis … and ending at line 316 “…a new nomenclature has been proposed to resolve this problem  [94] contains general information. The authors should make the manuscript more succinct and concise.

Response: In accord with this suggestion, we deleted general information in this manuscript as much as possible. (line 134, 291, 312, 334)

Comment 6: Table 2. List of interactions between RBPs and miRNAs in NDs.The Table is overloaded with data and hard to read. It should be either split or truncated.

 Response: In accord with this suggestion, we split the table by each disease. (Table 2-5)

Comment 7: Lines 624-625: “Lewy bodies, which are eosinophilic neuronal inclusions that contain both α-synuclein and ubiquitin, are pathological hallmarks of PD.”The authors should add here a reference on a recent review: Emamzadeh  et al. “Parkinson’s disease: Biomarkers, Treatment, and Risk Factors. Front. Neurosci., 018, 12:612 | https://doi.org/10.3389/fnins.2018.00612.

Response: In accord with this suggestion, we added your suggested article as a reference. (line 765)

Comment 8: Line 714 “…aberrant expression and/or misregulation of RBPs may one of the primary factors for miRNAs dysfunction”. This is incomplete sentence. Presumably, the authors want to say :” …aberrant expression and/or misregulation of RBPs may be one of the primary factors for miRNAs dysfunction”.

Response: Thank you for pointing this out. We have changed the sentence as you pointed out (line 875).